# CRISPR/Cas9 as a New Antiviral Strategy for Treating Hepatitis Viral Infections

**DOI:** 10.3390/ijms25010334

**Published:** 2023-12-26

**Authors:** Ulyana I. Bartosh, Anton S. Dome, Natalya V. Zhukova, Polina E. Karitskaya, Grigory A. Stepanov

**Affiliations:** The Institute of Chemical Biology and Fundamental Medicine, Siberian Branch, Russian Academy of Sciences, Novosibirsk 630090, Russia; uly-96@yandex.ru (U.I.B.); domeanton@ya.ru (A.S.D.); eschenko96@gmail.com (N.V.Z.); p.karitskaya@g.nsu.ru (P.E.K.)

**Keywords:** CRISPR/Cas, hepatitis, HBV, HCV, HAV, replication

## Abstract

Hepatitis is an inflammatory liver disease primarily caused by hepatitis A (HAV), B (HBV), C (HCV), D (HDV), and E (HEV) viruses. The chronic forms of hepatitis resulting from HBV and HCV infections can progress to cirrhosis or hepatocellular carcinoma (HCC), while acute hepatitis can lead to acute liver failure, sometimes resulting in fatality. Viral hepatitis was responsible for over 1 million reported deaths annually. The treatment of hepatitis caused by viral infections currently involves the use of interferon-α (IFN-α), nucleoside inhibitors, and reverse transcriptase inhibitors (for HBV). However, these methods do not always lead to a complete cure for viral infections, and chronic forms of the disease pose significant treatment challenges. These facts underscore the urgent need to explore novel drug developments for the treatment of viral hepatitis. The discovery of the CRISPR/Cas9 system and the subsequent development of various modifications of this system have represented a groundbreaking advance in the quest for innovative strategies in the treatment of viral infections. This technology enables the targeted disruption of specific regions of the genome of infectious agents or the direct manipulation of cellular factors involved in viral replication by introducing a double-strand DNA break, which is targeted by guide RNA (spacer). This review provides a comprehensive summary of our current knowledge regarding the application of the CRISPR/Cas system in the regulation of viral infections caused by HAV, HBV, and HCV. It also highlights new strategies for drug development aimed at addressing both acute and chronic forms of viral hepatitis.

## 1. Introduction

Hepatitis is an anthroponotic inflammatory liver disease, mainly induced by hepatitis viruses A (HAV), B (HBV), C (HCV), D (HDV), and E (HEV). Chronic hepatitis caused by HBV and HCV can lead to cirrhosis or liver cancer (hepatocellular carcinoma, HCC), while acute hepatitis can provoke potentially lethal acute liver failure. The World Health Organization (WHO) reports that approximately 296 million people worldwide suffered chronic HBV infection in 2019 and 58 million people had chronic HCV infection, with viral hepatitis causing more than a million deaths annually [1,2,3,4]. Although there are effective vaccines against HBV and HAV, the prevention of HCV and HEV infection remains a significant global health concern. Currently, interferon-α (IFN-α), nucleoside inhibitors, and reverse transcriptase inhibitors (for HBV) are used to treat viral hepatitis [5,6,7,8]. However, some of these approaches can only provide a partial cure for viral infections, and addressing their chronic forms is often challenging. Overall, the literature highlights the importance of finding new therapeutic agents against viral hepatitis.

The CRISPR/Cas9 system and its engineered modifications revolutionized the search for new treatment strategies, enabling the destruction of specific viral genome regions or influencing host-cell factors crucial for replication of the infectious agents [9]. This review summarizes our understanding of the application of CRISPR/Cas in regulating viral infections caused by HAV, HBV, HCV, and HEV. Additionally, it outlines new strategies for developing therapies against acute and chronic forms of viral hepatitis.

## 2. CRISPR Technology

In modern genetic engineering, the Clustered Regularly Interspaced Short Palindromic Repeats (CRISPR)/CRISPR-associated (Cas) system, particularly CRISPR/Cas9, has been a significant focus. This system includes short palindromic repeat clusters (30–40 bp), known as Clustered Regularly Interspaced Short Palindromic Repeats (CRISPR), associated with spacers (20–80 bp) corresponding to foreign genetic sequences. Functioning as a template for non-coding RNA (crRNA), the locus also contains genes encoding Cas9 proteins, serving as nucleases that induce precise double-strand breaks [10].

The CRISPR system’s arrays found in prokaryotes and archaea [11,12,13] play a role in immune memory mechanisms, with individual spacers protecting against infections through Watson-Crick base-pairing interactions [14]. The CRISPR/Cas9 system operates as adaptive immunity, involving complementary binding of foreign sequences to non-coding RNA, followed by degradation through Cas9 proteins. This versatile system finds extensive application in gene editing across various organisms, with implications for biological research, biotechnological product development, and disease treatment.

The CRISPR/Cas system is classified into six types [15] based on the sequence and structure of the Cas proteins. CRISPR/Cas type II, represented by a single Cas9 protein containing the HNH nuclease domain and RuvC-like nuclease domain, operates through mature CRISPR RNA (crRNA) [16] binding with trans-activating RNA (tracrRNA) [17]. The resulting tracrRNA/crRNA complex guides the Cas9 endonuclease to the target site, where a specific DNA sequence is cleaved by the Cas9 protein, mediated by a crRNA corresponding to the protospacer and a short motif adjacent to the protospacer known as the protospacer adjacent motif (PAM) [16,17]. Notably, the CRISPR type II system from Streptococcus pyogenes, involving Cas9 nuclease and chimeric synthetic single guide RNA (sgRNA or gRNA) [10], has become widely utilized, emphasizing the continuous evolution and diverse applications of genetic editing tools in modern research (Figure 1) [18,19].

Post-double-strand DNA break, cellular DNA repair processes commence through homologous recombination (HR) or non-homologous end joining (NHEJ). While NHEJ-mediated DNA repair rapidly restores DNA structure, it may generate insertions and deletions at target sites, potentially disrupting the function of target genes or genomic elements [20]. Therefore, careful control is required when using CRISPR/Cas9 to ensure the accuracy of the knockout of a specific DNA fragment [10,20,21].

## 3. Hepatitis B Virus

HBV is a human pathogen primarily targeting liver cells—hepatocytes. The immune-mediated damage of hepatocytes results in cirrhosis and hepatocellular carcinoma. In 2019, 296 million people worldwide were chronically infected with HBV [2]. HBV vaccination can reduce the incidence of HBV infection. However, the existing treatment can only weaken hepatitis infection, as it fails to eliminate the covalently closed circular DNA (cccDNA) that underlies chronic infection [6].

The modern classification assigns HBV to the Orthohepadnavirus genus of the Hepadnaviridae family [22]. The virion is a 42 nm infectious particle that consists of an outer envelope carrying the surface antigen HBsAg [23] and an inner nucleocapsid, which encloses the viral genome and the HBV DNA polymerase [24].

The HBV genome is a partially double-stranded circular DNA, 3200 bp in length (Figure 2). It contains four open reading frames (ORFs) that overlap completely or partially. The ORFs encode all viral regulatory elements: four promoters (core, S1, S2, and X), two enhancers (Enh1, Enh2), and signals of replication (DR1, DR2), polyadenylation, and encapsidation (ε) [25,26].

The HBV life cycle (Figure 3) involves the conversion of relaxed circular double-stranded DNA (rcDNA) into cccDNA in the hepatocyte nucleus. This cccDNA persists in the infected host cell as a stable nuclear minichromosome, serving as a template for the transcription of all viral proteins [27]. HBV replication begins with the synthesis of a negative-sense (−) DNA strand via reverse transcription of an intermediate RNA template called pregenomic RNA (pgRNA). Next, a positive-sense (+) DNA strand is formed, creating the final product of HBV replication—rcDNA. Viral mRNAs encode seven proteins: preCore (HBe precursor), core (HBc), protein X (HBx), envelope proteins (HBs L, M, and S), and DNA polymerase (P) [28]. The synthesis of viral nucleic acids is modulated by various cellular factors, but the lack of reliable cellular models for HBV infection and replication limits our understanding of these processes.

### 3.1. Targeting the CRISPR/Cas System to the HBV Genome

At present, the HBV infection treatment relies on using nucleoside and nucleotide analogs to inhibit reverse transcriptase [29]. However, this type of therapy cannot eliminate HBV infection due to the persistence of stable cccDNA [30]. One of the most effective tools to suppress cccDNA is the CRISPR/Cas9 system. It can directly disrupt various regions of the HBV genome, thus reducing the level of cccDNA [31,32,33,34,35,36] and the number of chromosomally integrated HBV sequences in infected cells [37]. The antiviral effect of this approach has also been demonstrated in vivo using hydrodynamic injection in mice [32,38,39]. Suppression of HbsAg expression through the CRISPR/Cas9-induced knockout of the preS1/preS2/S ORF can help reduce the malignant potential of HBV, including its proliferation in vitro and carcinogenicity in vivo [40], which could allow for preventing HCC that develops due to chronic HBV infection [41].

The effect of CRISPR/Cas9 can be enhanced by targeting it to HBV cccDNA. This is achieved, for example, by modulating DNA double-strand break repair pathways. Thus, inhibition of HDR and NHEJ by RI-1 or NU7026, respectively, can enhance the CRISPR/Cas9-induced reduction of HBV cccDNA levels [42]. Other studies utilizing CRISPR/Cas9 report that NU7026 prevents the degradation of HBV cccDNA [43]. These data demonstrate the importance of repair pathways in HBV cccDNA suppression. A synergistic effect of CRISPR/Cas9 and RNA interference can enhance the CRISPR/Cas9 function as well. HBV replication can be strongly inhibited through the simultaneous expression of two guide RNAs (gRNA) and a microRNA using a triple cassette (gRNA-microRNA-gRNA) [44].

Various modifications of the CRISPR/Cas9 system can also serve as tools for regulating HBV replication. For instance, a Cas9 nickase expressed with two gRNAs suppresses HBV replication in vitro and cleaves the HBV genome in mouse liver, whereas expression of a “dead” Cas9 (d-Cas9) with gRNA restrains HBV replication in vitro without disrupting the HBV genome structure. This indicates that utilizing Cas9 and d-Cas9 nickases with a pair of gRNAs can eliminate HBV DNA from the liver of patients with chronic hepatitis B [45].

One potential challenge with CRISPR/Cas9 targeting is the presence of cccDNA in various epigenetic forms. The methylation of cccDNA can produce heterochromatic patches, reducing the accessibility of chromatin to the targeted CRISPR/Cas9. The experimental data indicates that cccDNA methylation can significantly decrease the enzymatic activity of Cas9, but this effect is overcome by increasing the ratio of the CRISPR/Cas9 complex to the target cccDNA [46].

Thus, the literature data suggest that CRISPR/Cas9-mediated disruption of the HBV genome is a potent antiviral strategy against chronic hepatitis B. Nevertheless, eliminating multiple cccDNA copies from infected hepatocytes likely requires combining several therapeutic agents that target not only viral replication but also host-cell factors.

### 3.2. Host Factors Mediating HBV Infection

The lack of reliable models to culture HBV limits our understanding of the significance of host-cell factors at different stages of the HBV life cycle. Nonetheless, several studies have identified potential target genes mediating HBV infection, and knocking out these genes may suppress the infection and prevent HCC.

#### 3.2.1. Apolipoprotein E

Human apolipoprotein E (apoE) is produced by hepatocytes and plays a central role in the transport, metabolism, and homeostasis of cholesterol and other lipids [47]. CRISPR/Cas9-mediated knockout of the apoE-encoding gene leads to a >90% reduction in HBV infection in the HepG2^NTCP^ cell line and a >80% reduction in HBV production in HepAD38. However, apoE knockout has no significant impact on HBV cccDNA replication or the production of non-enveloped nucleocapsids in HepAD38 cells. Hence, inhibiting apoE, or interfering with its biogenesis, secretion, and/or binding to receptors, seems like a promising antiviral strategy to reduce the spread of HBV infection [48].

#### 3.2.2. Cellular Kinases

For efficient replication and transcription, HBV interacts with DNA damage response (DDR) pathways [49]. DDR has been shown to increase the activity of HBV cccDNA promoters both in vitro and in vivo [50]. DDR triggers the activation of several kinases, including PIKKs, ATM, ATR, and DNA-PKc [51]. Transcriptional activation of ATM or ATR kinases using CRISPR augments HBV replication. Furthermore, DNA damage results in the reactivation of HBV replication via upregulation of ATM and ATR [52]. Therefore, addressing the DDR pathways that affect HBV replication presents another strategy in drug discovery.

#### 3.2.3. Cellular Endonucleases

The HBV life cycle involves the formation of cccDNA from rcDNA. Unlike cccDNA, the rcDNA precursor contains a terminal redundancy^®^, which is a P protein-related sequence located at th’ 5’ end of the (−) DNA strand; rcDNA also includes a short RNA oligomer at th’ 5’ end of its (+) strand [28]. The P protein and the RNA oligomer are cleaved from the corresponding 5′ ends during the conversion of rcDNA into cccDNA [53,54]. However, the r sequence and the RNA oligomer can create a 5’-flap structure in rcDNA. It was shown that the endonuclease FEN1 can bind and cleave this 5′-flap structure of HBV rcDNA in vitro to facilitate its conversion to cccDNA. FEN1 knockout by CRISPR/Cas9 reduces cccDNA levels but does not alter cytoplasmic rcDNA in Hep38.7-Tet cells. This implies a direct involvement of FEN1 in HBV cccDNA formation [55].

#### 3.2.4. Host Antiviral Factors

APOBEC3A and APOBEC3B are antiviral effector proteins mediating the degradation of HBV cccDNA [56]. CRISPR/Cas9 knockout of the *APOBEC3A* gene in HepG2-1.1 cells increases HBV cccDNA levels by two-fold but has no significant influence on HBV transcription. Moreover, exposing the *APOBEC3B* promoter to CRISPR/Cas9 triggers both a twofold increase in HBV transcription and an approximately threefold increase in HBV cccDNA in HepG2-1.5 cells [57]. DNMT3A is another antiviral host factor. It methylates episomal HBV cccDNA, suppressing its transcription [58]. At the same time, DNMT3A overexpression reduces APOBEC3A in actHepG1-1.1 cells and APOBEC3B in HepG2-1.5 cells [57]. Overall, regulating the expression of antiviral factors is a potential strategy to combat HBV infection.

#### 3.2.5. Non-Coding RNAs

PCNA (proliferating cell nuclear antigen) plays multiple roles in DNA replication and repair as a coordinator of DNA polymerase [59]. At least four PCNA pseudogenes are suggested to exist: PCNAP1, PCNAP2, PCNAP3, and PCNAP4 [60]. Upregulated *PCNAP1* and *PCNA* are observed in the liver of HBV-infected chimeric mice, and elevated levels of PCNAP1 and PCNA mRNA are detected in the liver of patients with HBV, including those with HCC. CRISPR/Cas9-mediated knockout of *PCNA* reduces HBV cccDNA levels in HepG2 and HepG2.2.15 cells. In turn, PCNAP1 can enhance PCNA through sponging miR-154 targeting PCNA mRNA 3′UTR. Mediated by the interaction of PCNA with HBV cccDNA, this promotes HBV replication and cccDNA accumulation, as well as accelerated HCC growth in vitro and in vivo [61]. In addition, the established role of well-studied lncRNAs, such as MALAT1, HOTAIR, and others, in HBV replication and progression to associated disease states is noteworthy [62].

#### 3.2.6. Cellular Polymerases Essential for HBV Replication

To generate cccDNA from rcDNA, the viral DNA polymerase must be removed from th’ 5’ end of the minus DNA strand, and the RNA oligomer from th’ 5’ end of the plus strand needs to be cleaved. The cccDNA formation is finalized by synthesizing the plus DNA strand, trimming both strands’ ends, and ligating them. To complete the synthesis of the plus DNA strand, DNA polymerases are required. Yet, the formation of cccDNA during de novo infection does not require the viral DNA polymerase, as has been shown in HBV-infected HepG2^NTCP^ cells [63].

A key cellular factor essential for cccDNA formation is DNA polymerase kappa (Pol kappa); knocking out the *POLK* gene encoding this polymerase prevents the conversion of rcDNA to cccDNA. Similarly, the knockout of *POLL*, which encodes DNA polymerase lambda (Pol λ), can also lower the level of cccDNA, albeit to a lesser extent compared to the *POLK* knockout [63]. Another study established that DNA polymerases delta (Pol δ) and alpha (Pol α) are also necessary for cccDNA synthesis: CRISPR/Cas9-mediated knockout of *POLD1* encoding Pol δ1 prevented cccDNA formation in HepAD38 cells [64].

### 3.3. Host Factors Mediating HCC Development during HBV Infection

HBV infection can become chronic, which is associated with a high risk of liver cirrhosis, HCC, and increased lethality. The overall HCC incidence rate reaches 0.02% per 100 person-years in inactive carriers, 0.3% in subjects with chronic HBV without cirrhosis, and 2.2% in those with compensated cirrhosis [65]. There is considerable evidence that inhibition of HBV DNA replication can prevent hepatic fibrosis and reduce the incidence of HCC [66,67]. However, if left untreated, 15–25% of chronically infected people with HBsAg will develop HCC during the lifetime of infection [65]. The precise mechanisms of virus-mediated pathology are not fully understood. Recent studies using CRISPR/Cas shed light on the cellular factors that contribute to the development of HCC during HBV infection. Understanding these molecular pathways could help reduce the risk of developing HCC associated with chronic disease.

One such cellular factor is G9a, responsible for the demethylation of histone H3 lysine 9 (H3K9). G9a upregulation can promote the development of HCC [68]. G9a participates in the development, differentiation, and regulation of immune cells, thus modulating inflammatory responses [69]. CRISPR/Cas9 knockout of G9a suppresses HCC cell proliferation and migration in vitro and inhibits HCC tumorigenicity in vivo, suggesting that targeted application of CRISPR/Cas9 can halt HCC development both in vitro and in vivo [70].

Other factors influencing the development of HCC are cellular receptors. For example, high expression of CXC chemokine receptor type 4 (CXCR4) is linked to HCC invasion, progression, and metastasis [71]. In contrast, CRISPR/Cas9-mediated knockout of CXCR4 inhibits proliferation, migration, and invasion of HepG2 cells while also reversing epithelial-mesenchymal transition (EMT), increasing HCC chemosensitivity, and reducing its malignancy in vitro and in vivo [72]. Nuclear receptor coactivator 5 (NCOA5) also plays an important role in HCC development. Knockout of NCOA5 using CRISPR/Cas9 in HCC cells inhibits their proliferation and tumor microsphere formation by suppressing EMT [73].

In addition to receptors, aspartate-β-hydroxylase has a significant impact on malignant transformation. The development of HCC slows down if the *ASPH* gene encoding NCOA5 is knocked out by CRISPR/Cas9 [74]. Another unfavorable factor in chronic HBV infection is the increased expression of *WNT3A*, which produces the Wnt-3a protein involved in the Wnt/β-catenin signaling pathway. Its activation contributes to the highest risk of developing HCC among chronic HBV carriers [75]. In HepG2 cells, CRISPR/Cas9-mediated knockout of *WNT3A* prevents proliferation and colony formation due to deactivation of the Wnt/β-catenin pathway and cell cycle arrest in the G1 phase. Additionally, *WNT3A* can suppress xenograft tumor growth in vivo [76]. The role of p53 and PTEN in HCC development has also been demonstrated. Mutations in *TP53* and dysregulated expression of *PTEN* are the most frequently occurring events in HBV-associated HCC [77,78]. When delivered by a hydrodynamic tail vein injection, CRISPR/Cas9 can target the tumor suppressor genes *p53* and *Pten* in the liver of HBV transgenic mice. The CRISPR/Cas9-induced mutations in *p53* and *Pten* accelerate hepatocarcinogenesis in adult mice [79].

Some viral proteins can also activate signaling cascades that provoke HCC. For example, hepatitis B virus X protein (HBx) is involved in hepatocarcinogenesis and considered oncogenic [80,81]. CDC42 protein overexpression has also been reported in various cancers, including HBV-associated HCC [82]. CRISPR/Cas9-mediated knockout of CDC42 significantly reduces HBx-promoted proliferation and inhibits apoptosis in HBx-stimulated HuH-7 cells (HuH-7-HBx). It also suppresses IQGAP1, a downstream effector of CDC42. Thus, CDC42 knockout in HuH-7-HBx cells results in the suppression of the entire HBx/CD42/IQGAP1 signaling pathway that can promote carcinogenesis [83].

Non-coding RNAs contribute to the development of HCC as well. MicroRNAs have been shown to modulate key cellular processes underlying carcinogenesis [84]. To illustrate, miR-3188 knockout by CRISPR/Cas9 suppresses cell growth, migration, and invasion in HCC cell lines and inhibits xenograft tumor growth in mice. The same study demonstrates that the HBx–miR-3188–ZHX2-Notch1 signaling pathway plays an important role in the pathogenesis and progression of HBV-related HCC [85].

Since HCC is directly linked to high levels of HBsAg in patients [41], identifying the factors crucial for HBsAg expression is an important therapeutic strategy. Hyrina et al. performed a genome-wide CRISPR screen and identified multiple host factors required for HBsAg expression. ZCCHC14, TENT4B, and TENT4A were shown to stabilize HBsAg expression through a nontemplated addition to the 3′ end of the HBV RNA, a process called RNA tailing. This mechanism is dependent on the HBV post-transcriptional regulatory element (PRE) and is inhibited by RG7834, which destabilizes HBV transcripts [86,87]. These data pave the way for potential therapeutic strategies against HBV-related HCC [88].

### 3.4. Modification of T Lymphocytes as a Strategy to Suppress HBV Infection

HBV antigens are processed and presented by major histocompatibility complex (MHC) molecules on the surface of infected cells [89]. HBV-specific T cells can interact with peptides presented through leukocyte histocompatibility antigen (HLA), alleviating antiviral and tumor burdens [90]. Yet, the HBV-specific T-cell response may weaken during chronic HBV infection [91]. In a recent study, HBV-specific T cells were engineered to express a recombinant T cell receptor (rTCR) while downregulating endogenous T cell receptors (eTCR). This was facilitated by the novel CRISPR-STOP technology, which introduced premature stop codons into the homologous regions of invariant TCR chains 1 and 2 beta (TRBC 1/2) [92,93]. Disruption of eTCR was coupled with a higher expression of the introduced rTCR, making engineered T cells enriched with rTCR. Thus, pre-manufactured banks of eTCR-/rTCR+ T cells could be successfully used to treat HBV infection [94,95].

However, certain T-cell factors may interfere with the control of chronic HBV infection. One of them is the programmed death factor-1 (PD-1). By binding to its ligand PD-L1, it blocks the activation of the PI3K pathway and suppresses the proliferation and differentiation of T lymphocytes [96]. PD-1 and targeted therapy against HBV can act in synergy, suppressing HBV gene expression and significantly increasing the survival of transgenic HBV mice [97].

### 3.5. Epigenetic Editing of cccDNA

The CRISPR/Cas system with catalytically inactive, “dead” Cas9 (dCas9) can be exploited for the epigenetic editing of cccDNA. Fusions of dCas9 with epigenetic effector domains can induce the rewriting of epigenetic marks at multiple target locations in the viral genome. Hence, this method can be utilized in the future to regulate the HBV genome without introducing double-strand breaks [98,99].

## 4. Hepatitis C Virus

Like HBV, HCV is a human pathogen that infects liver cells and, in some cases, causes fibrosis, cirrhosis, and hepatocellular carcinoma. According to WHO statistics, 58 million people were chronically infected with HCV in 2019 [3]. To date, no HCV vaccine has been approved. Available therapies, including IFN-α and nucleoside inhibitors, are not universally effective [5,8] and have significant side effects [8]. The only targeted drugs currently prescribed for chronic hepatitis C are daclatasvir [100], an inhibitor of the viral protein NS5A, and sofosbuvir [101], an inhibitor of NS5B. They are often used in combination with each other or with other drugs [102]. However, the high cost of this treatment and its significant side effects encourage researchers to seek alternative therapeutic approaches.

A member of the *Hepacivirus* genus and the *Flaviviridae* family [103], HCV is a viral particle with a diameter of 50–80 nm. The virion has an outer lipid shell (envelope) with anchored glycoproteins E1 and E2. Inside the envelope, the core protein forms a nucleocapsid containing the viral genome [104]. The HCV genome is a 9.6-kb single-stranded positive sense (+) RNA that encodes a single polyprotein of approximately 3000 amino acids. The HCV polyprotein is cleaved by viral and host-encoded proteases into three structural proteins (core, E1, and E2), the hydrophobic peptide p7, and six non-structural (NS) proteins (NS2, NS3, NS4A, NS4B, NS5A, and NS5B) [105] (Figure 4).

The HCV life cycle (Figure 5) involves the interaction of multiple host-cell factors [106]. Therefore, a better understanding of the relationship between HCV and host proteins may shed light on potential targets for controlling chronic HCV infection.

### 4.1. Targeting the CRISPR/Cas System to the HCV Genome

Despite breakthroughs in the development of direct-acting antivirals (DAAs) targeting viral proteins NS3, NS4A, NS5A, and NS5B [107], these drugs often induce desensitization or resistance. This is associated with viral replication errors, introducing polymorphisms in genes that encode proteins targeted by DAAs [108,109]. Therefore, targeting highly conserved regions of the HCV genome is considered the most promising treatment option. For instance, in a recent study, CRISPR/Cas13 aimed at the highly conserved internal-ribosomal entry site (IRES) of HCV RNA which decreased HCV replication and viral protein translation in the Huh-7.5 cell line [110].

Apart from CRISPR/Cas13, the Cas9 system from the gram-negative bacterium *Francisella novicida* (FnCas9) can target HCV RNA. This system is also able to inhibit HCV RNA replication and viral protein translation. In the near future, FnCas9 is likely to become one of the most promising systems for combating HCV infection [111].

### 4.2. Host Factors Mediating HCV Infection

#### 4.2.1. Factors Enabling HCV Entry into Cells

HCV requires attachment and post-attachment receptors to enter cells. Essential HCV attachment receptors on hepatocytes include TIM-1 receptors (or HAVCR1) [112], SDC-1 and SDC-2, and heparan sulfate proteoglycan receptors (HSPG) that enable ApoE-mediated HCV binding [113]. Knockouts of SDC-1 and SDC-2 by CRISPR/Cas9 lead to a significant reduction in HCV infection [114].

Post-attachment receptors required for HCV entry include OCLN, CLDN1, and CD81 [115,116,117,118]. CRISPR/Cas9-mediated knockout of the OCLN gene suppresses infection of the human liver cell line Huh7.5.1-8 with various HCV genotypes [119]. Similarly, knockout of CLDN1 in Huh-7.5 cells promoted their resistance to infection with the HCV genotype 2a Jc1. HCV of all genotypes can use CLDN1 to enter the cell, although some viruses can also interact with CLDN6 and CLDN9 [120]. LDN1/CLDN6 double-knockout Huh-7.5 cells support infection by a mutant HCV only when CLDN1, CLDN6, or CLDN9 is expressed [121].

The CD81 receptor is not only important for HCV infection of cells but also plays the role of a cellular factor influencing the infectivity of HCV. CRISPR/Cas9-mediated CD81 knockout in HEK 293T cells can increase or restore the infectivity of HCV pseudo-particles (HCVpp) [122].

The post-attachment receptors CD81, CLDN1, OCLN, and LDLR have also been found essential for cell-to-cell transmission of infection. Knockout of the *CLDN1* and *OCLN* genes reduces cell-to-cell transmission of HCV by 60–80% [114,119], while microRNAs complementary to *CLDN1*, *OCLN*, *CD81*, and *LDLR* promote a more than 50% reduction in HCV transmission. Unlike post-attachment receptors, the attachment receptors SDC-1, SDC-2, and TIM-1 play only a minor role in cell-to-cell transmission. Knockout of the *SDC-1*, *SDC-2*, or *TIM-1* genes results in an approximately 20% reduction in cell-to-cell HCV transmission [114].

#### 4.2.2. SPCS1

A genome-wide CRISPR/Cas9-based screen helped identify host genes that result in reduced flavivirus infection when knocked out. A subset of ER-associated signal peptidase complex (SPCS) proteins was shown to be required for proper cleavage of the flavivirus structural proteins and secretion of viral particles. Loss of *SPCS1* expression led to a marked decrease in the E2 glycoprotein and a weaker HCV infection [123].

#### 4.2.3. FKBP6

Viruses often rely on chaperones for their replication. For instance, many viruses, particularly HCV, use Hsp90 [124]. HCV replication has also been shown to require the FKBP6 co-chaperone. CRISPR/Cas9-mediated FKBP6 knockout results in a complete suppression of HCV replication in hepatoma cell lines [125].

#### 4.2.4. Non-Coding RNAs

Long non-coding RNAs (lncRNAs) are important regulators of cellular processes. LncRNAs modulate transcriptional or post-transcriptional mechanisms [126] and can also control viral infection and immune responses. For example, lncRNA can regulate *IFI6*, a member of the interferon-stimulated gene (ISG) family. An IFN-induced lncRNA-IFI6 negatively regulates *IFI6* expression through histone modification of its promoter, thereby increasing HCV replication. Targeting CRISPR/Cas9 to the sequence encoding this lncRNA, located within the *IFI6* gene, suppresses HCV infection [127]. A recent study [128] has introduced the lncRNAs LINC00152 and UCA1 as diagnostic and prognostic markers for HCV-induced hepatocellular carcinoma.

#### 4.2.5. RTN3

Exosomes play a crucial role in the development of HCV infection by facilitating the transfer of infectious material from one infected cell to another. Infectious viral material detected within exosomes is characteristic of hepatitis C [129]. Reticulons (RTNs) are shown to be directly associated with the formation of infectious exosomes. For instance, RTN3 mediates the specific loading of viral exosomes, which are then released by infected cells. CRISPR/Cas9-mediated knockdown of RTN3 isoforms L and S significantly lowers the number of infectious exosomes released by Huh7 cells [130].

#### 4.2.6. p38α

Several stages of the HCV life cycle require p38α, one of the four representatives of p38 mitogen-activated protein kinases (p38 MAPK), as a crucial cellular factor. CRISPR/Cas9-mediated p38α-knockdown (without monoclonal isolation) significantly reduces HCV core protein levels, as well as intracellular and extracellular HCV RNA, in the Huh7.5.1 cell line. In this case, HCV triggers TAB1-dependent activation of p38α, as knockdown of TAB1 using CRISPR/Cas9 impairs p38α phosphorylation induced by HCV infection. Thus, the downregulation of p38α activation pathways appears to be a promising antiviral strategy [131].

#### 4.2.7. P-Body Protein Mov10

Mov10 is a processing body (P-body) protein that, alongside other cellular factors, mediates HCV infection. It has been demonstrated that overexpression or CRISPR/Cas-mediated knockout of Mov10 suppresses intracellular HCV RNA levels in Huh7.5.1 cells and also reduces the infectivity of the released virus. Thus, both excess and depletion of Mov10 can decrease HCV replication and infection levels [132].

#### 4.2.8. Host Antiviral Factors

Interferons (IFNs) and interferon-stimulated genes (ISGs), play a central role in antiviral responses. For example, *IFI6* is required to suppress HCV infection. CRISPR/Cas9-mediated knockout of *IFI6* significantly elevates HCV RNA and core protein levels in Huh7.5.1 cells infected with HCV, genotype 2a JFH1 [127]. Another essential antiviral factor in HCV infection is interferon-stimulated gene 15 (*ISG15*). In U2OS cells, HCV RNA level increases when *ISG15* is knocked out using CRISPR with a double-mutant Cas9 nickase D10A (Cas9n) [133]. This indicates that *ISG15* is involved in suppressing HCV RNA replication [134].

STAT1 and STAT2, mediators of the interferon-dependent pathway, have also been demonstrated to be essential for signaling in response to IFN-α (STAT2) and IFN-λ (STAT1, STAT2). When stimulated with IFN-λ and infected with HCVcc (HCV grown in cell culture), the cell line Huh-7.5 produces less NS5A and viral RNA, while PKR, STAT1, and STAT2 are upregulated. This effect of IFN-λ is prevented by STAT1 or STAT2 knockout. Similarly, in a STAT2 knockout cell line, neither NS5A nor viral RNA levels decrease, and there is no PKR upregulation upon IFN-α stimulation and HCVcc infection. This suggests that STAT2 is required for inhibiting HCV replication via not only the IFN-λ pathway but also through IFN-α-dependent signaling [135].

Another antiviral factor is RNA-activated protein kinase (PKR) which is involved in the innate immune response to HCV. When activated by dsRNA intermediates during HCV genome replication, PKR also promotes inhibition of the p53 tumor suppressor, which can cause HCC [136]. In the HepG2 cell line expressing miR-122 [137], CRISPR/Cas9 knockout of PKR increases the level of HCV core protein, suggesting PKR-mediated suppression of HCV replication. It has also been shown that PKR knockout restores p53 accumulation and p53-mediated responses to DNA damage [138]. Thus, these data demonstrate the dual role of PKR: contributing to the innate immune response on the one hand and promoting HCC on the other.

However, HCV is capable of suppressing innate immune response signaling in infected cells. For instance, it can activate the E3 ubiquitin ligase PDLIM2, which stimulates the nuclear degradation of STAT2 required for IFN-λ and IFN-α signaling [135]. CRISPR/Cas9-mediated knockout of PDLIM2 in the Huh7.5 cell line provokes STAT2 upregulation and its accumulation in the nucleus following IFN-α stimulation. This results in reduced levels of HCV core protein and intracellular HCV RNA, as well as lower HCV production during HCV infection. Notably, more virus is required to infect PDLIM2 knockout cells compared to the naive Huh7.5 cell line [139].

#### 4.2.9. PKLR

Pyruvate plays a key role in the regulation of HCV replication. It is produced during glycolysis when pyruvate kinase catalyzes the conversion of phosphoenolpyruvate to pyruvate. CRISPR/Cas9 knockdown of the pyruvate kinase gene *PKLR* leads to a decrease in HCV RNA and core protein levels in Huh7.5.1 cells infected with the genotype 2a JFH1 HCV. The same study reports that *PKLR* expression is regulated by the non-coding RNA miR-130a; CRISPR/Cas9-mediated knockdown of miR-130a results in an upregulation of *PKLR* and HCV core protein [140]. Inhibiting PKLR could be a potential strategy to mitigate HCV infection.

#### 4.2.10. CRISPR Screening of Cellular Factors Essential for HCV

A CRISPR screen, using lentiGuide-Puro from the GeCKO v2 library at an MOI of 0.3, showed that the HCV JFH1-resistant Huh7.5.1 cell population was highly enriched in guide RNAs (gRNAs) targeting the HCV receptors CD81, OCLN, and CLDN1, which confirms their role in HCV entry [115]. Another significantly enriched gene was *ELAVL1*. Also referred to as HuR, ELAVL1 is an RNA-binding protein involved in mRNA stabilization [141]. In Huh7.5.1 cells with an *ELAVL1* knockout, HCV RNA replication was practically inhibited. In addition, the CRISPR screen identified that HCV replication requires microRNA-122 and the DGCR8 protein processing it [115]. Other essential HCV replication factors include the enzymes RFK and FLAD1. They are involved in the two-step conversion of riboflavin to flavin adenine dinucleotide (FAD). Both *RFK*- and *FLAD1*-knockout cells were resistant to HCV replication. That said, FAD, or exogenous flavin mononucleotide (FMN), rescued HCV replication in *RFK*-knockout cells, whereas FAD but not FMN rescued viral replication in *FLAD1*-knockout cells. This indicates that HCV replication is dependent solely on sufficient FAD levels [115].

## 5. Hepatitis A Virus

HAV is a human pathogen that infects liver cells and is the primary cause of acute hepatitis. In severe cases, it can lead to acute liver failure and death. WHO reports that in 2016, more than 7000 people died from hepatitis A worldwide [1]. To date, there is no specific treatment for HAV infection, although vaccines are available to prevent it [142].

HAV belongs to the Picornaviridae family, genus Hepatovirus [143]. It is a small, non-enveloped virus measuring 27 nm that can be cloaked in host-derived membranes, forming 50-110 nm virions [144]. HAV has an icosahedral protein capsid consisting of 60 copies of each of its four major structural proteins: VP1, VP2, VP3, and VP4 [145].

The HAV genome (Figure 6) is a single-stranded (+) RNA, approximately 7.5 kb in length, containing a long ORF that encodes a single polyprotein. The viral protease 3Cpro processes the polyprotein into ten mature proteins [146].

### 5.1. Host Factors Mediating HAV Infection

#### 5.1.1. Factors Enabling HAV Entry into Cells

Hepatitis A virus cellular receptor 1 (HAVCR1) is a functional cellular receptor for HAV. Disrupting the monkey ortholog of HAVCR1 through CRISPR/Cas9 knockout leads to a loss of susceptibility to HAV infection in AGMK cells. The cells acquire resistance to free viral particles (vpHAV) and exosomes purified from HAV-infected cells (exo-HAV). Conversely, transfection of HAVCR1 cDNA or its mouse ortholog (mHavcr1) into HAVCR1 knockout cells restores their susceptibility to vpHAV and exo-HAV infection [147]. HAVCR1 knockout in the Huh7 cell line, however, results in resistance to exo-HAV, but not vpHAV, suggesting the presence of an alternative HAV receptor in Huh7 cells [148].

Moreover, it is not only the HAVCR1 receptor but also the cholesterol transporter NPC1 that participates in cargo delivery from exo-HAV into the cytoplasm. These two receptors facilitate clathrin-mediated endocytosis and interact in the late endosome. CRISPR/Cas9 knockout of *HAVCR1* and *NPC1* in the Huh7 cell line demonstrates that both receptors are necessary for the membrane fusion and delivery of viral RNA from exo-HAV into the cytoplasm [148].

#### 5.1.2. Factors of the Innate Immune Response

Like HCV, HAV can suppress the innate immune response through the activation of PDLIM2. During HAV infection in Huh7.5 cells, CRISPR/Cas9-mediated knockout of the *PDLIM2* gene leads to a downregulation of the HAV nuclear protein and a decrease in HAV RNA, both in cells and the supernatant [139].

#### 5.1.3. GRP78

The endoplasmic reticulum (ER) is essential for viral replication. However, viruses can induce ER stress, triggering the unfolded protein response (UPR) [149,150]. UPR signaling is governed by glucose-regulated protein 78 (GRP78), an ER chaperone. GRP78 interacts with three downstream mediators: PERK, ATF6, and IRE1. In response to ER stress, they reduce the levels of newly synthesized proteins translocated into the ER lumen, enhance ER protein-folding capacity and secretion potential, and facilitate the transport and degradation of ER-localized proteins [151]. In Huh7 cells, CRISPR/Cas9-mediated knockout of the *HSPA5* gene encoding GRP78 promotes increased HAV replication and downregulation of other ER stress molecules, including ATF4, ATF6, and XBP1 [152].

## 6. Hepatitis E Virus

HEV is the most common cause of enterically transmitted viral hepatitis. HEV infection can lead to acute self-limiting hepatitis and, occasionally, to potentially fatal fulminant hepatitis (acute liver failure). According to the WHO, approximately 20 million cases of HEV infection are reported annually, and hepatitis E caused around 44,000 deaths in 2015 [4]. To date, only one HEV vaccine is available, approved solely in China [153]. There are also no specialized drugs for HEV infection and chronic cases are treated with nucleoside inhibitors [7].

HEV belongs to the Orthohepevirus genus of the Hepeviridae family, which includes four species (Orthohepevirus A-D). Species A is subdivided into eight genotypes. Genotypes 1 and 2 (HEV-1 and HEV-2) can only infect humans, whereas genotypes 3 and 4 (HEV-3 and HEV-4) have been described in both humans and animals [154].

The HEV virion is a small 32–34 nm particle with an icosahedral capsid comprised of 180 copies of the capsid protein. Structurally, there are three functional domains: the S domain forms the icosahedral shell, which serves as the base for the M and P domains. The P domain is hypothesized to be the binding site for the cellular receptor.

The HEV genome (Figure 7) is a single-stranded (+) RNA of approximately 7.2 kb in length, containing three major ORFs (ORF1, ORF2, and ORF3). Genomic (+) RNA functions as mRNA for translation of the viral nonstructural polyprotein from ORF1. The capsid protein ORF2 and the multifunctional protein ORF3 are translated from the bicistronic subgenomic mRNA generated from (−) RNA [155].

### BISPR

Transcriptome analysis of the Huh7 cell line upon HEV transfection revealed upregulation of the *BST2* gene and the lncRNA BISPR regulating it [156]. BST2, or tetherin, is an IFN-stimulated factor that interferes with the budding of some enveloped viruses by anchoring virions to the cell membrane [157]. CRISPR/Cas9-mediated knockout of *BISPR* in Huh7 cells results in an eightfold increase in HEV release 24 h after HEV replicon transfection. Such cell lines can hold great potential in the development of an attenuated vaccine against HEV [158].

## 7. Conclusions

The CRISPR/Cas9 system and its modifications help deepen our understanding of the mechanisms behind the development of viral infections, especially hepatitis. By directing CRISPR/Cas to both the viral genome and host factors that support viral replication, we can potentially mitigate or eliminate the infection. The development of vaccine formulations and model cell lines can also greatly benefit from engineered CRISPR/Cas systems targeting cellular factors that enhance viral replication. As described in this review, various examples of CRISPR/Cas applications in regulating HAV, HBV, HCV, and HEV infections unveil novel strategies for developing therapeutic agents against acute and chronic viral hepatitis.

## Figures and Tables

**Figure 1 ijms-25-00334-f001:**
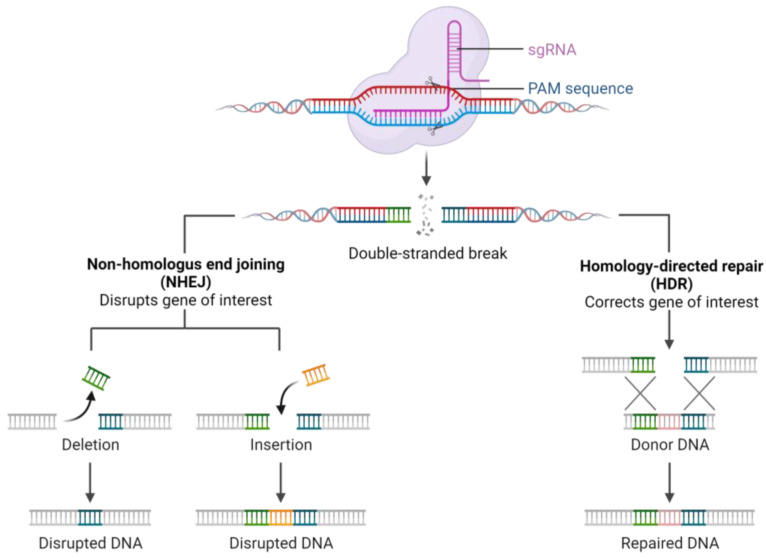
Mechanism of CRISPR/Cas9 system action.

**Figure 2 ijms-25-00334-f002:**
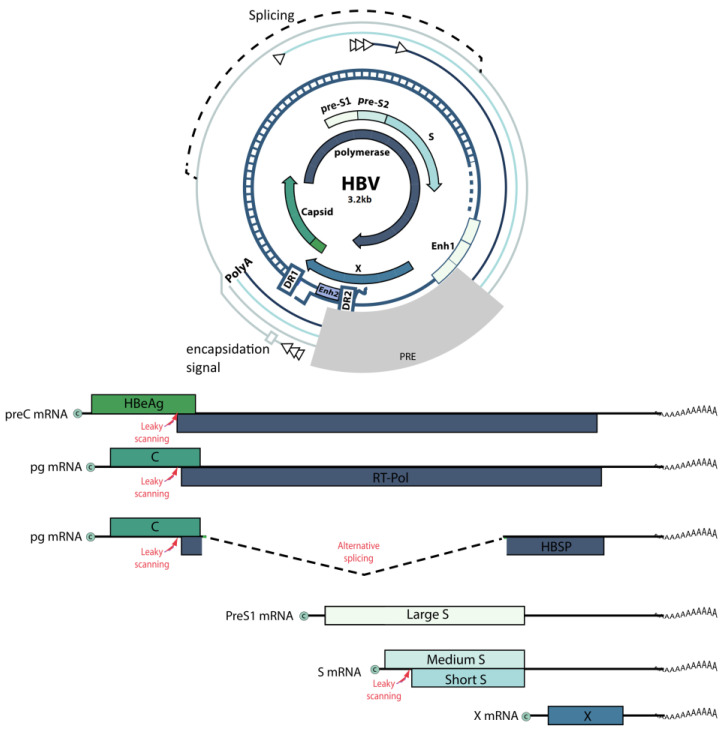
Schematic representation of the HBV genome.

**Figure 3 ijms-25-00334-f003:**
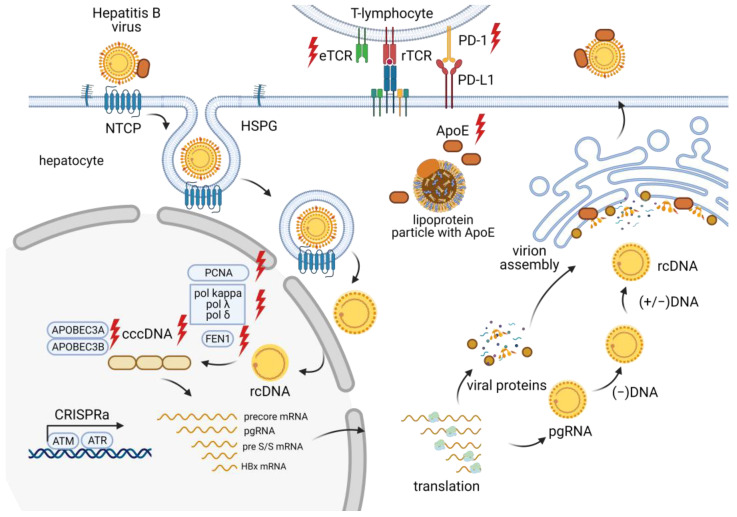
Overview of the CRISPR/Cas effects at different stages of the HBV life cycle.

**Figure 4 ijms-25-00334-f004:**
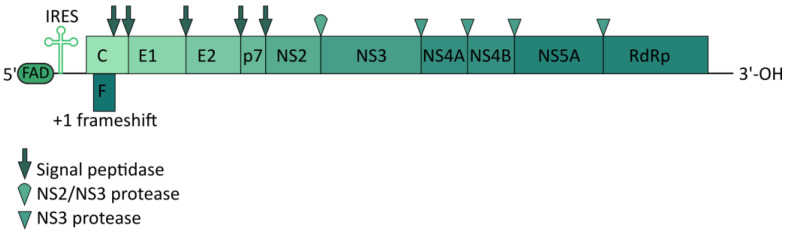
Schematic representation of the HCV genome.

**Figure 5 ijms-25-00334-f005:**
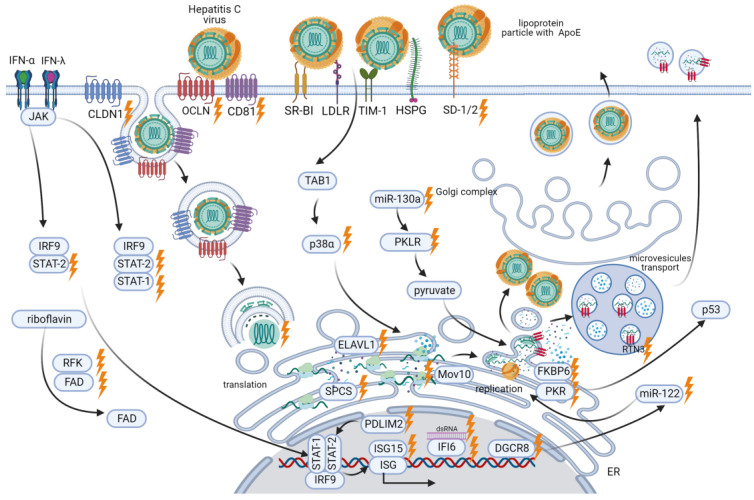
Overview of CRISPR/Cas applications at different stages of the HCV life cycle.

**Figure 6 ijms-25-00334-f006:**
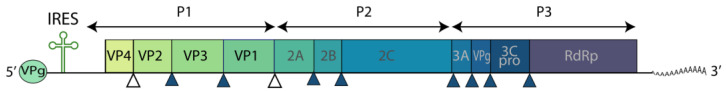
Schematic representation of the HAV genome.

**Figure 7 ijms-25-00334-f007:**
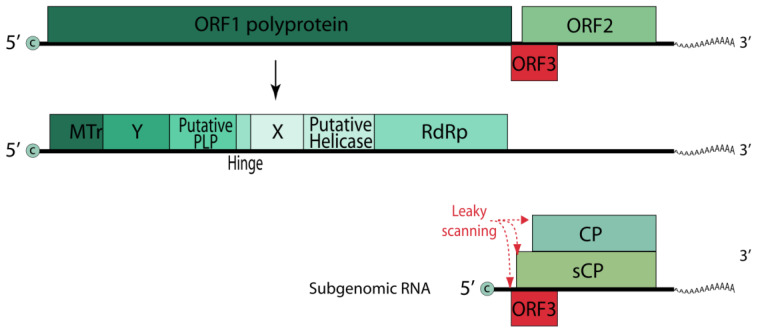
Schematic representation of the HEV genome.

## Data Availability

Data is contained within the article.

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
