# Peer review of "CRISPR/Cas9 as a New Antiviral Strategy for Treating Hepatitis Viral Infections"

_ijms, 2023, doi:10.3390/ijms25010334_

Round 1
Reviewer 1 Report
Comments and Suggestions for Authors
This is a review article so there are no novel findings here which is fine. There are only 2 figures in the article. Figures help the reader understand better what is going on than huge blocks of words. I encourage the authors to add an additional figure or two describing the genome structure of the hepatitis virus and perhaps add a phylogenetic tree so that we can see how the different viruses are related to each other on a tree.
In addition, there is only a tiny paragraph describing exosomes which will be of great interest to many readers. I encourage the authors to significantly expand the paragraph about exosomes and provide more detail on this subject.
Author Response
Thank you very much for taking the time to review this manuscript. Please find the detailed responses below in the attachment and the corresponding revisions/corrections in track changes in the re-submitted files.

Reviewer 2 Report
Comments and Suggestions for Authors
The review written by Bartish et al., entitled "CRISPR/Cas in the control of viral hepatitis: novel treatment strategies" is globally well presented and it is considered with a big interest as the new technology of CRISP/Cas is interesting to be use as antiviral new strategy.
Review is easy to understand, Figures are well done and very representatives of the content of the text. However,, some rrevisions are needed to be taken in consideration to improve the quality of the review.
Major revision:
* In a new independant section instead of Introduction section, Authors should describe well the technology of CRISPR: Discovery, history, fundamental aspect using references...and add a new Figure descibing the fundamental function of this technology in genomes.
* A new title is suggested to most describe the review: "CRISP/Cas9, as a new antiviral strategy for treating hepatitis viral infections".
* Abstract should be improved, authors should reduce sentences regarding hepatitis pathologies and replace them by sentences talking about the technology of CRISP/Cas: principle, efficacity and opportunities of using...
Minor revision:
* Please eliminate references in Abstract section
* Abstract: pp28, eliminate HEV, it is not concerned by the review
* Keywords: Please add HAV also, correct hepatits by hepatitis and specify by CRISPR/Cas
* References in all the text should be used regularly, ie pp41 [1-4] and pp44 [5-8] and in many other sentences...
Author Response

(The authors gave the same response as above.)

Round 2
Reviewer 2 Report
Comments and Suggestions for Authors
Revised version is suitable for publication